# A defined mechanistic correlate of protection against *Plasmodium falciparum* malaria in non-human primates

Alexander D. Douglas [1], G. Christian Baldeviano [2], Jing Jin[1], Kazutoyo Miura[3], Ababacar Diouf[3], Zenon A. Zenonos[4], Julio A. Ventocilla[2], Sarah E. Silk[1], Jennifer M. Marshall[1], Daniel G.W. Alanine[1], Chuan Wang[1], Nick J. Edwards [1], Karina P. Leiva[2], Luis A. Gomez-Puerta [2], Carmen M. Lucas[2], Gavin J. Wright[4], Carole A. Long[3], Joseph M. Royal[2] & Simon J. Draper [1]

Malaria vaccine design and prioritization has been hindered by the lack of a mechanistic correlate of protection. We previously demonstrated a strong association between protection and merozoite-neutralizing antibody responses following vaccination of non-human primates against *Plasmodium falciparum* reticulocyte binding protein homolog 5 (PfRH5). Here, we test the mechanism of protection. Using mutant human IgG1 Fc regions engineered not to engage complement or FcR-dependent effector mechanisms, we produce merozoite-neutralizing and non-neutralizing anti-PfRH5 chimeric monoclonal antibodies (mAbs) and perform a passive transfer-*P. falciparum* challenge study in *Aotus nancymaae* monkeys. At the highest dose tested, 6/6 animals given the neutralizing PfRH5-binding mAb c2AC7 survive the challenge without treatment, compared to 0/6 animals given non-neutralizing PfRH5-binding mAb c4BA7 and 0/6 animals given an isotype control mAb. Our results address the controversy regarding whether merozoite-neutralizing antibody can cause protection against *P. falciparum* blood-stage infections, and highlight the quantitative challenge of achieving such protection.

[1] Jenner Institute, University of Oxford, Old Road Campus Research Building, Roosevelt Drive, Oxford OX3 7DQ, UK. [2] US Naval Medical Research Unit No. 6 (NAMRU-6), Av. Venezuela Cuadra 36, Bellavista, Callao, Peru. [3] Laboratory of Malaria and Vector Research, NIAID/NIH, 12735 Twinbrook Parkway, Rockville, MD 20852, USA. [4] Cell Surface Signalling Laboratory, Wellcome Trust Sanger Institute, Cambridge CB10 1SA, UK. Correspondence and requests for materials should be addressed to A.D.D. (email: sandy.douglas@ndm.ox.ac.uk) or to S.J.D. (email: simon.draper@ndm.ox.ac.uk)

To our knowledge, immunological protection against the disease-causing blood-stage of the major human malaria parasite *Plasmodium falciparum* has never been proven to be caused by any single specific immunological mechanism. Numerous immune effectors have been proposed as potential mediators of both naturally acquired and vaccine-induced protection against the *P. falciparum* blood-stage: merozoite neutralization independent of FcR bearing cells (either with or without complement)[1,2]; inhibition of parasite multiplication by soluble factors released by monocytes in response to merozoite binding by antibody, so-called antibody-dependent cellular inhibition (ADCI)[3]; neutrophil antibody-dependent respiratory burst activity upon encounter with opsonized merozoites[4]; merozoite phagocytosis by monocytes or neutrophils[5,6]; multiple effects of antibody binding to the infected erythrocyte surface[7]; and inhibition of blood-stage parasites by mechanisms orchestrated by CD4+ or CD8+ T cells[8,9]. Analysis of associations between activity measured in these assays and naturally acquired clinical immunity are complicated by the extent of exposure to malaria. Seminal passive transfer experiments in humans involved the transfer of total human IgG with multiple specificities and multiple potential effector mechanisms[10,11].

The best-standardized and most widely used tool for assessment of blood-stage vaccine-induced antibody functionality is the assay of growth inhibitory activity (GIA). This measures antibody-mediated *P. falciparum* merozoite neutralization in erythrocyte cultures lacking any FcR-bearing cells and without addition of complement[1]. Notably, the relationship between GIA and naturally acquired malaria immunity is contentious[12], with the latter very likely due to a complex interplay of multiple immune responses. In contrast, an association between GIA and vaccine-induced protection has been observed in four non-human primate (NHP) vaccination studies[13–16]. Presence of additional unmeasured vaccine-induced causes of protection could not be excluded in these studies.

In order to resolve this uncertainty and to provide the field with clarity regarding appropriate means of selecting improved anti-merozoite vaccines, we wished to test the existence of a causal relationship between GIA-inducing antibody and protection against *P. falciparum*. We therefore set out to perform a passive transfer study in *Aotus nancymaae*, i.e., intact immunocompetent NHP which are highly susceptible to the *P. falciparum* blood-stage parasite. We demonstrate that high levels of neutralizing antibodies can achieve protection. This demonstration of a causal relationship between a defined immunological mechanism and protection may enable the use of GIA as a mechanistic correlate of protection[17] after vaccination of humans, facilitating future vaccine development.

## Results

**Properties of chimeric anti-PfRH5 mAbs.** We previously reported the isolation of a panel of murine mAbs against PfRH5[18]. 2AC7 is the most potent merozoite-neutralizing anti-PfRH5 mAb of which we are aware: it recognizes a conformational epitope which has not been accurately mapped but, on the basis of competition binding experiments, is thought to be similar to that of 9AD4 (another mAb) at the apex of PfRH5's kite-like structure[18,19]. 4BA7 is a non-neutralizing anti-PfRH5 mAb which binds a linear epitope in a disordered loop remote from the 9AD4 and basigin binding sites[18,19]. To achieve sufficient in vivo half-life of these two mAbs for an NHP passive transfer study, we developed mouse-human IgG1 chimeric 2AC7 (c2AC7), and chimeric 4BA7 (c4BA7). To exclude the possibility of Fc-mediated antibody effector functions, the Fc region used was a previously reported version (hIgG1Δnab) with mutations in

regions critical for binding to the activating human FcRs (FcγRIa, FcγRIIa and FcγRIIIa) and complement component C1q[20]. This mutant Fc region fails to activate monocytes, human natural killer cell-mediated antibody-dependent cellular cytotoxicity, or cell lysis via the classical complement pathway[20] (and personal communication from M. Clark and K. Armour). As a control, we produced versions of an irrelevant human mAb, EBL040 (directed against the Ebola virus glycoprotein) with two Fc regions, both hIgG1Δnab and wild-type hIgG1. All mAbs referred to in this report carried the hIgG1Δnab except where use of EBL040 with the wild-type hIgG1 Fc is explicitly stated.

To assess their suitability for use in our planned study (and particularly in view of the fact that there is, to our knowledge, no prior data regarding use of human mAbs in *Aotus spp.*), we characterized the resulting mAbs in vitro and in vivo. c2AC7 and c4BA7 behaved similarly to their parental murine mAbs in the assay of GIA, with an EC$_{50}$ of 12 μg/mL for c2AC7 (Fig. 1a). The hIgG1Δnab Fc region abrogated interaction of mAbs with *A. nancymaae* C1q (as assessed by ELISA, Fig. 1b) and FcγRIa, FcγRIIa and FcγRIIIa (as assessed by surface plasmon resonance [SPR], Fig. 1c, d). In a pilot pharmacokinetic study, in vivo half-life of the c4BA7 mAb plasma concentration was 7 days (Fig. 1e).

**Protection against *P. falciparum* by neutralizing PfRH5 mAb.** We proceeded to perform two studies in which *Aotus* received a challenge with 10,000 *P. falciparum* FVO-strain blood-stage parasites, followed by doses of mAb. Initial doses were administered immediately after challenge; quantities of these initial doses are those referred to subsequently as the dose level for each group. Informed by the pharmacokinetic data, "top-ups" of half the original dose were given at days 7 and 14 to maintain steady plasma concentrations. We initially tested a high dose of 100 mg/kg, with the aim of providing a definitive answer about mAb-mediated protection. At this dose, the GIA-inducing mAb c2AC7 was highly effective in suppressing parasitemia whereas the non-GIA-inducing anti-PfRH5 c4BA7 mAb had no effect, similar to the irrelevant EBL040 mAb (Fig. 2a–c). As previously used, the protocol-specified primary endpoint for efficacy analyses was log$_{10}$-transformed cumulative parasitemia up to the day on which the first animal within the study reached a treatment endpoint (Fig. 2d; henceforth "LCP")[14,21]. At day 10, when the first animals in the EBL040 and c4BA7 groups required treatment, parasitemia was not microscopically detectable in animals receiving 100 mg/kg c2AC7 ($P < 0.0001$ for reduction in LCP by c2AC7 as compared to EBL040 by the protocol-specified one-tailed unpaired t-test with Welch's correction for unequal variance; $P = 0.002$ by two-tailed Mann–Whitney test; $P > 0.2$ for difference between LCP in c4BA7 and EBL040 recipients by either analysis). Late in the challenge, from day 29 onward, low-level parasitemia was detected microscopically in some c2AC7-recipient animals: repeatedly in 2/6 animals and on a single occasion in a further 2/6 animals, while the remaining 2/6 animals were microscopically negative throughout (Fig. 2c). At the end of the challenge period (using larger blood samples which could only be obtained on a single occasion), parasitemia was detected in 3/6 animals using a sensitive quantitative polymerase chain reaction (qPCR) assay with a lower limit of detection of approximately 20 parasites/mL of blood[22]; the remaining 3/6 animals were qPCR negative (Fig. 2e).

**Relationship between mAb concentration and protection.** In a separate challenge study, we assessed the effect of lower doses of c2AC7 (Fig. 2f–h). Protection was not analyzed statistically in view of the small number of animals at each dose level ($n = 2$). Although there was a trend towards delayed treatment in some

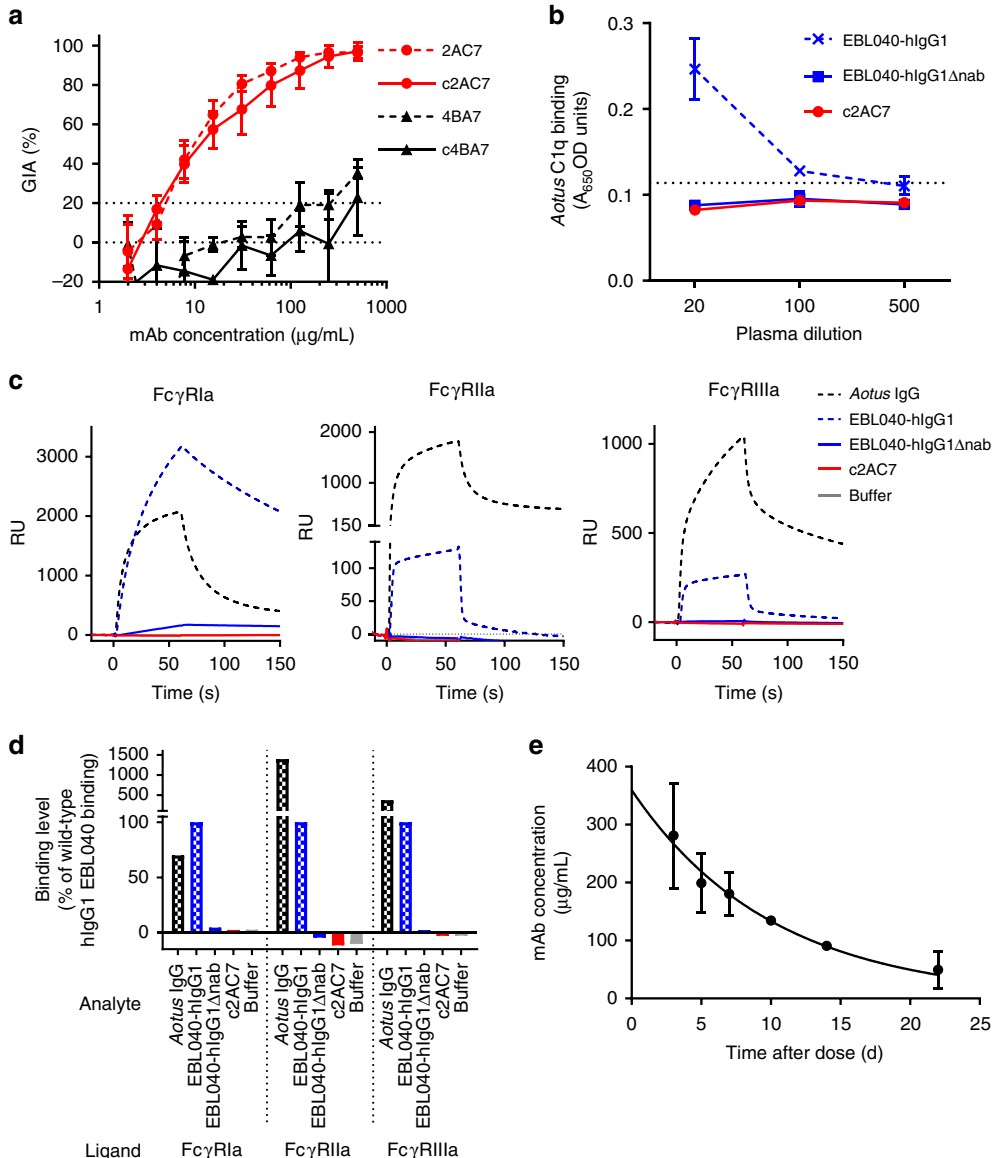

**Fig. 1** Chimeric mAbs have desired properties for a challenge study. Panel **a** shows GIA with murine (dashed lines) and chimeric (solid lines) versions of 2AC7 (red lines) and 4BA7 (black lines) against FVO-strain parasites. Error bars indicate range of two independent experiments (with each experiment's datum being the mean of three replicate wells); lines connect the mean of the experiments. Results of −20 to 20% GIA are regarded as negative. Panel **b** shows absence of ELISA-detectable interaction of *A. nancymaae* C1q with mAbs bearing the hIgG1Δnab Fc. Horizontal dotted line indicates background (mean plus two standard deviations of control wells coated with mAbs but not receiving plasma); points and error bars indicate mean and range of two replicate wells per condition. Panels **c** and **d** show SPR data demonstrating that the hIgG1Δnab Fc abrogates binding to *A. nancymaae* Fcγ receptors. Panel **c** shows reference-subtracted (Fc2-1) binding-response curves for the injection of various antibody analytes (indicated by line characteristics, as shown in the legend to the right) over chips with FcγRIa (left), FcγRIIa (middle), and FcγRIIIa (right) ligands captured on Fc2; in each case, analyte is injected from 0 to 60 s. Panel **d** summarizes peak (60 s) binding of each antibody to each receptor, expressed as percentage of the response obtained on the same receptor with the wild-type human IgG1 mAb control (EBL040-hIgG1). Panel **e** shows results of preliminary pharmacokinetic study: anti-PfRH5 ELISA-measured plasma mAb concentrations from days 3 to 22 after administration of a single dose of 30 mg/kg c4BA7 to two *A. nancymaae*. Points are mean of two animals' results; error bars show the range, although are not visible where replicates were in close agreement. Line shown is the fitted one-phase exponential decay curve

animals receiving c2AC7 at doses from 8 to 33 mg/kg, all animals developed sustained parasitemia and ultimately required treatment. Despite the need to exercise care in comparing between separately performed challenge studies, the outcomes in these animals were clearly different to outcomes at the 100 mg/kg dose level (Fig. 2c vs. h).

To further explore the relationship between anti-PfRH5 antibody concentration, parasite neutralization and protection against *P. falciparum*, we performed ELISA and GIA assays.

ELISA-measured plasma concentrations of mAbs in the high-dose study are shown in Fig. 3a. The geometric mean c2AC7 concentration in animals receiving 100 mg/kg was 728 µg/mL (range: 626–914 µg/mL) at day 3, and similar concentrations were maintained until around day 21 (due to the use of top-up doses). As expected, plasma concentrations of mAbs in the low-dose study were proportionately lower (Fig. 3b). To assess relationships between mAb concentration and outcome, we calculated time-weighted mean [TWM] mAb concentrations for each

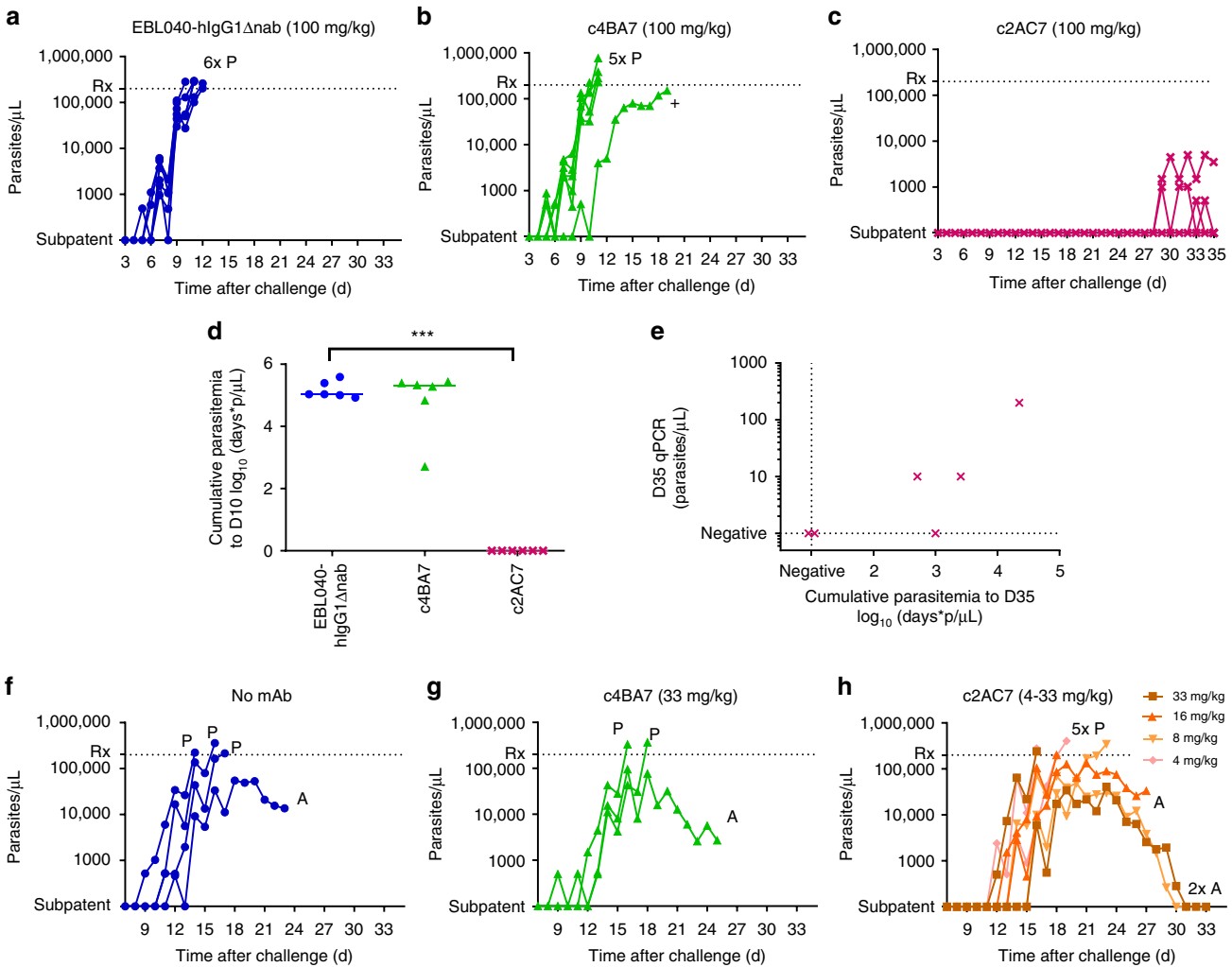

**Fig. 2** A GIA-inducing, but not a non-GIA-inducing, anti-PfRH5 mAb protects against blood-stage *P. falciparum*. Panels **a–c** show timecourse of parasitemia in animals receiving EBL040-hIgG1Δnab (panel **a**), c4BA7 (panel **b**), or c2AC7 (panel **c**), each at 100 mg/kg body weight ($n = 6$ per group). Upper horizontal dashed line indicates the 200,000 parasites/μL (p/μL) threshold for initiation of antimalarial treatment (Rx) because of hyperparasitemia. "P" indicates treatment of an animal due to hyperparasitemia; "+" indicates a single animal found dead on day 19; occasional unexpected deaths have previously been recorded among *Aotus* both before and during *P. falciparum* challenge[14]. Panel **d** shows cumulative parasitemia up to day 10 (on which the first animal in the study reached a treatment endpoint) as per the prespecified primary analysis. Individual datapoints and median are shown. ***$P < 0.0001$ by one-tailed *t* test with Welch's correction. Panel **e** shows cumulative microscopically-detected parasitemia up to day 35 (the end of the study) vs. qPCR-measured parasitemia on day 35 in the c2AC7 group. Two animals had no detectable parasitemia by either method. Panels **f–h** show timecourse of parasitemia in the lower-dose study, annotated as for panels **a–c**; "A" indicates treatment due to anemia. Panel **f** shows animals receiving no mAb ($n = 4$); panel **g** shows animals receiving c4BA7 ($n = 3$) at 33 mg/kg; panel **h** shows animals receiving c2AC7 at a range of doses from 33 mg/kg down to 4 mg/kg, as indicated by the legend ($n = 2$ per dose level)

animal for the period days 0–14 (providing a single representative concentration measure, despite the fluctuation in mAb concentrations over time due to top-up dosing). There was a relationship between TWM c2AC7 concentration and challenge outcome (Fig. 3c). The lowest TWM c2AC7 concentration in a 100 mg/kg c2AC7 recipient was 614 μg/mL; this animal had no detectable parasitemia at any point in the study by microscopy or qPCR. Time-weighted mean c2AC7 concentrations were as high as 185 μg/mL in the low-dose study in which all animals required treatment.

Consistent with the measured pharmacokinetics, antibody concentrations fell fairly rapidly after the final dose (Fig. 3a, b, d). By day 35, c2AC7 concentrations in some 100 mg/kg recipients were in the same range as peak concentrations in the (non-protected) 33 mg/kg recipients: the two 100 mg/kg recipients with repeated microscopically detectable parasitemia after day 28 were

those with the lowest c2AC7 concentration by day 35 (111 and 156 μg/mL; Fig. 3d). Thus although peak concentrations in the 100 mg/kg recipients were protective against emergence of patent parasitemia, subpatent parasitemia must have persisted; the artificially rapid late decline in antibody concentrations (faster than would be expected after vaccination or hIgG1 administration to humans[23]) unmasked such persistent parasites. Further extension of the study, with the frequent blood sampling necessary to assure prompt treatment if indicated, was not permitted for animal welfare reasons.

We previously reported a quantitative relationship between PfRH5 vaccine efficacy in *Aotus* and antibody activity in the assay of GIA[14]. We introduced the measure of "GIA$_{50}$ titer", calculated by dividing the plasma total IgG concentration by the total IgG GIA EC$_{50}$[14]—thus giving the plasma dilution factor required to achieve 50% GIA in the assay. We found that animals were

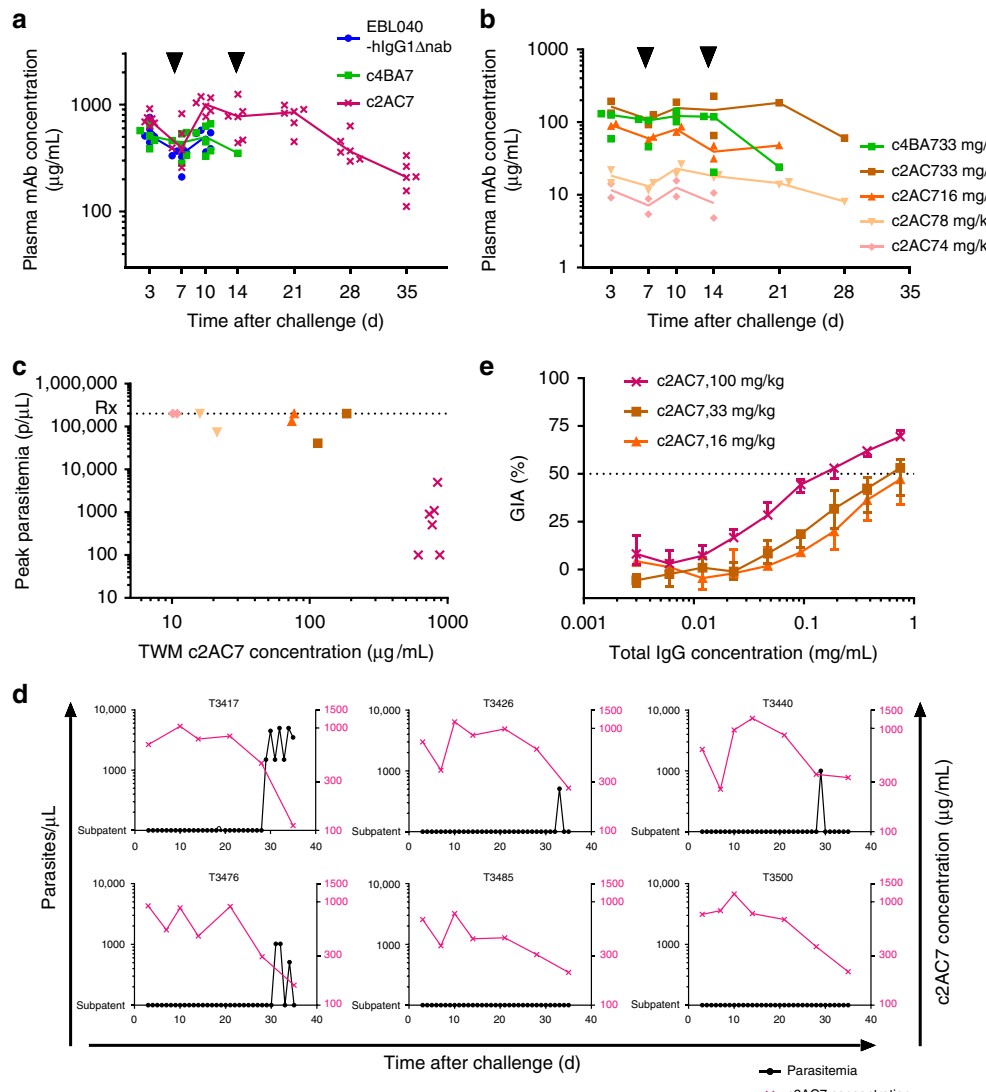

**Fig. 3** Plasma mAb concentrations, GIA and outcome. Panels **a** and **b** show ELISA-measured plasma mAb concentrations in the high-dose and lower-dosing studies, respectively, up to the point of reaching a treatment endpoint. Points indicate data from individual animals; lines link group medians. Arrowheads indicate mAb top-up dose administration on days 7 and 14. Panel **c** shows the relationship between time-weighted mean (TWM) c2AC7 mAb concentration and peak parasitemia in individual animals, combining data from the high-dose and lower-dosing studies (Spearman correlation coefficient $r_S = -0.8$, $P = 0.001$). Panel **d** shows timecourses of parasitemia (black circles, left axis) against ELISA-measured c2AC7 concentrations (pink crosses, right axis). Each subplot represents an individual animal from the 100 mg/kg c2AC7 group. Parasitemia data is as shown in panel 2**c** and ELISA data is as shown in panel 3**a**. Panel **e** shows the relationship between concentration of total protein G-purified IgG and GIA (against FVO parasites). Due to small sample volumes, samples from pairs of *Aotus* (collected on day 10 after challenge) were pooled prior to IgG purification. Results shown for the 100 mg/kg c2AC7 group ($n = 6$ animals) are the median (points) and range (error bars) of the three such pairs, with each datum being the median of three independent assays. Results for the 33 and 16 mg/kg c2AC7 groups ($n = 2$ animals each) are each from a single pair of animals, showing the median (points) and range (error bars) of three independent assays. All assays had triplicate wells at each tested IgG concentration. Results are not shown for animals from other groups, in which GIA did not reach 50% at the maximum tested concentration: GIA was <20% at 0.75 mg/mL of IgG from animals in the EBL040, c4BA7 and no mAb control groups, and <30% at 0.75 mg/mL for the 8 and 4 mg/kg c2AC7 groups

protected against a similar *P. falciparum* challenge if the GIA$_{50}$ titer exceeded 5; animals with GIA$_{50}$ titers as high as 3.5 were not protected. The comparison included animals receiving quite different PfRH5-based vaccines and additional unmeasured effectors could not be excluded. Here, because of ethical restrictions on blood sampling during the challenge resulting in small sample volumes, GIA was assessed using IgG purified from day 10 samples pooled from 2 monkeys within a single treatment group (Fig. 3e). The median GIA EC$_{50}$ for 100 mg/kg c2AC7 recipients was 0.16 mg/mL total IgG (range: 0.12–0.22 mg/mL); the GIA EC$_{50}$ for 33 mg/kg c2AC7 recipients was 0.61 mg/mL.

GIA$_{50}$ titers in the protected pairs of 100 mg/kg c2AC7 recipients ranged from 14 to 20; the GIA$_{50}$ titer for the non-protected pair of 33 mg/kg c2AC7 recipients was 4. The similarity of these results to those in our earlier vaccination study is consistent with the hypothesis that merozoite-neutralizing antibody, as measured by GIA, is the principal cause of protection in both contexts.

## Discussion

To our knowledge, this is the first example of protection of primates against *P. falciparum*, the principal cause of human malaria

mortality, by transfer of a mAb: indeed, as far as we are aware, no single immune effector—of known specificity, in the absence of others—has previously been shown to protect either humans or primates against *P. falciparum*.

We recently reported protection by an anti-PfRH5 mAb in an immunodeficient humanized mouse model engrafted with human hepatocytes and erythrocytes, but the dose–response relationship between mAb concentration, GIA and protection was not studied[24]. Moreover, the mice were receiving ongoing immuno-suppressive drug treatment, had only 20–30% human (and hence *P. falciparum* susceptible) erythrocytes at the onset of blood-stage infection, and had microscopically detectable circulating schizonts (suggesting that the model does not fully recapitulate the cytoadherence of late-stage parasites which is a critical feature of *P. falciparum* pathogenesis). Our present study complements and extends that finding in an intact immunocompetent NHP model known to support *P. falciparum* sequestration[25,26].

Protection was achieved here by a mAb with a mutant Fc designed to be incapable of engaging FcR or complement-mediated effector functions. The presence of an Fc region contributes to merozoite neutralization by 2AC7 and other anti-PfRH5 antibodies, perhaps via bivalent binding or steric effects, but independent of FcR[18]. Importantly, the objective of this study was to validate the possibility of protection by an FcR-independent mechanism (and hence the use of the GIA assay as a surrogate of protection): not to formally exclude the possibility of FcR-dependent augmentation of protection by anti-RH5, and certainly not to exclude the possibility of FcR-dependent protection by antibodies of other specificities. PfRH5's prior-itization as a vaccine antigen (and the isolation of the 2AC7 mAb) has been on the basis of the GIA assay and its essential interaction with the receptor basigin (suggesting the possibility of invasion inhibition by FcR-independent steric blockade)[18,27,28]. To our knowledge, PfRH5 has never been shown to be a target of FcR-dependent anti-merozoite effects and we are doubtful that it is an appropriate target. Rather, the principal targets of antibodies acting via such mechanisms appear to be abundant merozoite surface proteins[2,29,30]—perhaps a necessity for recruitment of these functions—while PfRH5 is neither abundant nor easily accessible on the merozoite surface until late in invasion[18,31].

There remains, in our opinion, a need for validation of the in vivo protective effects against malaria of complement- and FcR-dependent antibody effector mechanisms. Candidate mAbs selected specifically for their potency in in vitro assays of these activities (most likely binding to non-PfRH5 antigens, and ideally without potent neutralizing activity) could readily be tested for their in vivo protective capacity, comparing versions with wild-type Fc and with mutant Fc regions such as hIgG1Δnab. Such direct evidence is likely to provide stronger support for vaccine development than can be obtained from epidemiological studies, in which the multifactorial nature of natural immunity makes it impossible to isolate the causal contribution to protection of a single mechanism.

Levels of anti-PfRH5 antibody and GIA required to suppress parasite multiplication were high, despite the in vitro potency of c2AC7 (we are not aware of any mAb which is substantially more potent in the assay of GIA, either from our own work or that of others). Anti-PfRH5 concentrations necessary to achieve protection were orders of magnitude higher than those commonly seen in individuals from malaria-endemic regions: in a previous study testing sera from 175 Kenyan and Ghanaian adults, we found median anti-PfRH5 responses of <0.1 μg/mL and a maximum response of 3 μg/mL[32]. We therefore doubt that naturally-acquired anti-PfRH5 responses are major contributors to natural immunity.

Achievement of a given reduction in parasite multiplication in vivo (whether after vaccination[14] or here after passive transfer),

appears to require neutralizing antibody concentrations which are multiples of those needed to achieve similar effects in vitro. This is consistent with other pathogens, such as smallpox and polio, for which neutralization assays are accepted surrogates of protection[33]. The precise quantitative relationship between anti-PfRH5 antibody concentration, GIA and protection is likely to differ between malaria-naïve *Aotus* and humans living in malaria-endemic areas: current and future clinical trials will assess this relationship in humans, and the possibility of synergy between vaccine-induced anti-PfRH5 responses and naturally-acquired responses to other parasite antigens. Notwithstanding these possibilities, the attainment and maintenance in humans of the levels of antibody required to achieve protection in this study would be extremely challenging with currently available vaccine platforms.

Our data thus invite two interpretations. On a positive note, in a field which has struggled to understand the mechanisms of vaccine-induced protection, this is perhaps one of the clearest dissections of the quantitative relationship between any candidate *P. falciparum* vaccine effector, an in vitro assay, and in vivo protection. On the other hand, although merozoite-neutralizing antibody can in principle be protective, there is a substantial quantitative hurdle to be overcome in order to develop a clinically useful vaccine acting by this mechanism. This clarity is likely to be instructive for future vaccine candidate selection: the level of GIA induced by anti-PfRH5 antibodies should serve as a benchmark to be exceeded by future vaccines aiming to protect by merozoite neutralization, and the validation of additional possible mechanisms of protection should be a priority.

## Methods

**Monoclonal antibody production**. We previously described production of 2AC7 and 4BA7 anti-PfRH5 murine hybridomas[18]. mRNA isolated from the hybridomas was used as a template to RT-PCR amplify the respective antibody rearranged variable heavy and light chains[34]. Heavy- and light-chain variable region coding sequences were then gene synthesized (Thermo Fisher) and subcloned into transient mammalian expression vectors in frame with the hIgG1Δnab[20] or hIgCκ chain constant sequences as appropriate.

Isolation of the plasmids encoding the light and heavy chains of the human IgG1 monoclonal EBL040, directed against Ebola virus glycoprotein, has been reported elsewhere[35]. The EBL040 $V_H$ and $V_L$ sequences were subcloned using InFusion (Clontech) into mammalian expression vectors in frame with hIgG1Δnab[20], wild-type hIgG1 (to produce the EBL040 with wild-type hIgG1, used only in Fig. 1b–d) or hIgCκ chain constant sequences.

Monoclonal antibody (mAb) was produced by transient transfection of plasmids into HEK293F cells in Expi293 media (Thermo Fisher) using 25 kDa linear polyethylene-imine (PEI, Polysciences). mAb was purified on a protein G column, with a 0.5 M arginine wash step for endotoxin reduction, followed by a size-exclusion chromatography (SEC) step to remove aggregates and further reduce endotoxin, and concentration adjustment/buffer exchange using a centrifugal filter device (Vivaspin, Sartorius). SEC chromatograms confirmed >99.5% purity (in that >99.5% of the mAb preps were of the expected molecular weight). Endotoxin levels were <0.5 EU/mg (LAL Chromogenic Assay Kit, Thermo Fisher).

**Binding of mAbs to *Aotus nancymaae* C1q and Fcγ receptors**. C1q binding was assessed by ELISA. Maxisorp plates (ThermoFisher) were coated overnight with EBL040-hIgG1, EBL040-hIgG1Δnab, or c2AC7 mAbs (50 μL at 5 μg/mL in phosphate-buffered saline [PBS]), and blocked for 1 h with Blocker Casein (ThermoFisher). A dilution series of pooled plasma from mAb-naïve *A. nancymaae* (5-fold from a 1:20 starting dilution) was applied for 2 h, followed by application of horseradish-peroxidase-conjugated sheep anti-human C1q polyclonal antibody (Abcam ab46191, diluted 1:100 in Blocker Casein) for 1 h. Plates were developed with TMB substrate (ThermoFisher) for 1 h and absorbance was measured at 650 nm on a Clariostar reader (BMG).

FcγRIa, FcγRIIa, and FcγRIIIa were selected for study from the set of Fcγ receptors on the basis that these are the major 'activating' effector-function mediating Fcγ receptors in humans[36]. Predicted mRNA sequences encoding *A. nancymaae* homologs of these human receptors were identified by searching NCBI data arising from the Owl Monkey Genome Project (Baylor College of Medicine Human Genome Sequencing Centre; https://www.hgsc.bcm.edu/non-human-primates/owl-monkey-genome-project). Sequences retrieved (NCBI accession numbers XM_012442107.1, XM_012449596.1, XM_012449592 and the corresponding predicted protein sequences XP_012297530.1, XP_012305019.1 and XP_012305015.1) were BLAST

aligned against human receptor reference sequences (Uniprot P12314, P12318, P08637). Probable *Aotus* receptor ectodomain sequences were inferred from the alignments, corresponding to amino acids 54–345 of XP_012297530.1 for *Aotus* FcγRIa, amino acids 1–216 of XP_012305019.1 for FcγRIIa, and amino acids 103–310 of XP_012305015.1 for FcγRIIIa. In the cases of FcγRIa and FcγRIIIa, the first amino acids of the selected regions aligned with the human orthologs, while the predicted N-terminal portions of the *Aotus* protein sequences did not align with the N-terminus of the human orthologs and are likely to have represented erroneous bioinformatics predictions. The coding sequences for the predicted *Aotus* receptor ectodomains were synthesized with flanking NotI and AscI restriction enzyme sites (Thermo Fisher), and subcloned into a previously reported pTT3-based vector supplying C-terminal tag sequences (rat CD4 domains 3 and 4, biotin acceptor peptide, and hexa-histidine (6xHis))[37]. Biotinylated receptor protein was produced by transient co-transfection of HEK293F cells with these plasmids and a plasmid encoding the BirA biotin ligase[37]. Harvested cell supernatant was dialyzed extensively against PBS.

Surface plasmon resonance was performed using a Biacore X100 instrument and biotin-CAP chip (both from GE Healthcare), at an analysis temperature of 25 °C and a flow rate of 30 μL/min. Running buffer comprised HEPES-buffered saline, pH 7.4, supplemented with 3 mM EDTA, 0.05% Tween-20, 1 mg/mL salmon DNA and 2 mg/mL carboxymethyl-dextran (all from Sigma) HBS-EP+ running buffer[38]. At least 2500 response units (RU) of biotinylated FcγR protein was captured on flow cell 2 (Fc2). Fc1 received CAP reagent alone.

Analyte test samples comprised two positive controls for binding (protein G-purified polyclonal *Aotus* IgG and EBL040 mAb with wild-type human IgG1 Fc, EBL040-hIgG1); two mAbs with the mutant hIgG1Δnab Fc (EBL040-hIgG1Δnab and c2AC7), and blank (buffer only). All antibodies were tested at 100 μg/mL. Analytes were injected over the both flow cells for 60 s. Peak background subtracted binding (i.e., Fc2-1) was calculated.

**Animals and veterinary procedures.** The experiments reported herein were conducted in compliance with the Animal Welfare Act and in accordance with the principles set forth in ref. [39]. The study protocol was approved by NAMRU-6's Institutional Animal Care and Use Committee (protocol number NAMRU-6 14-06); the Department of the Navy Bureau of Medicine and Surgery (NRD-927); the University of Oxford's central committee on Animal Care and Ethical Review (ACER); and the Institut Nacional de Recursos Naturales (INRENA) at the Peruvian Ministry of Agriculture.

Adult female owl monkeys (*A. nancymaae*) were housed at the US Naval Medical Research Unit No. 6 (NAMRU-6), Lima, Peru. Intravenous (i.v.) administration of antibodies and parasites and withdrawal of blood samples from the saphenous vein was performed under ketamine anesthesia. EDTA-anticoagulated blood was prepared using standard methods to obtain plasma and in some cases PBMC.

In a preliminary study to assess mAb pharmacokinetics, $n = 2$ *Aotus* were administered a 30 mg/kg i.v. bolus of chimerized anti-PfRH5 mAb c4BA7. Plasma samples were collected at intervals up to day 22; mAb concentrations measured by ELISA (see below); and a single-phase exponential decay model was fitted to data collected from day 3 onward to estimate terminal elimination half-life. On day 7 of this preliminary study, one animal additionally received an anti-CD20 mAb (Arzerra, GSK), as part of a concurrent study to be published separately; exclusion of data collected after that administration altered the point estimate of the fitted half-life by less than 1 day.

For challenge studies, randomization to groups was stratified by prestudy weight and gender. Animals were challenged i.v. on day 0. Challenge inocula comprised $10^4$ FVO-strain *P. falciparum* iRBC taken from a donor monkey[40]. Immediately after administration of the challenge inoculum and a PBS flush, the appropriate dose of mAb was administered i.v. in 2 mL of PBS. Additional doses of mAb, each half the initial dose given to that animal, were administered on days 7 and 14. Antibodies and doses were as set out in figure legends; passively transferred EBL040 mAb also had the hIgG1Δnab Fc.

From day 3, daily thin-film parasitemia quantification and alternate-day Hct measurements were conducted. Animals were treated when (i) parasite density reached ≥200,000/μL; or (ii) Hct fell to ≤25% (severe anemia is a common complication of poorly controlled malaria infection in *Aotus*); or (iii) upon reaching challenge day 35 if no other treatment endpoint had been reached.

*P. falciparum* quantitative PCR was performed using a TaqMan-based assay[22], modified in that test samples comprised 200 μL of packed *Aotus* RBCs, filtered through CF11 cellulose powder to remove leukocytes[41].

**ELISA and GIA.** c2AC7, c4BA7, EBL040, and total *Aotus* IgG in plasma samples were quantified by ELISA.

In the case of c2AC7 and c4BA7, streptavidin-coated plates (ThermoFisher) were coated with saturating quantities of monobiotinylated PfRH5 protein produced by transient transfection of HEK293E cells (Canadian National Research Council)[14,27]. For EBL040, plates were coated with 100 ng per well of trimeric Ebola virus Zaire glycoprotein (aa 1–650) with a four amino acid C-terminal C-tag (EPEA)[42], produced by transient transfection of HEK293 cells[14], supernatant concentration by tangential flow filtration with 10 kDa membrane cut-off (Millipore), CaptureSelect C-tag affinity chromatography (Thermo Fisher Scientific) using 2 M MgCl₂, 20 mM Tris pH 7.4 elution buffer, and finally

SEC in Tris-buffered saline using a HiLoad Superdex16/600 200 pg column (GE Healthcare). All three antibodies were detected using alkaline-phosphatase-conjugated goat anti-human IgG (Sigma A3187, 1:1000 dilution).

For measurement of day 0 *Aotus* IgG (prior to mAb administration), a self-sandwich design was used: Maxisorp plates (ThermoFisher) were coated with rabbit anti-monkey IgG polyclonal antibody (Sigma A2054, 1:1000 dilution); the secondary antibody was an alkaline-phosphatase-conjugated version of the same antibody (Sigma A1929, 1:1000 dilution).

Plates were developed with para-nitrophenyl-phosphate substrate (Sigma) in diethanolamine buffer (ThermoFisher) and read using an ELx800 instrument (Biotek). Conversion to plasma concentration was made by comparison of OD readings to a dilution series of the appropriate mAb or protein G-purified *Aotus* IgG[43].

For Fig. 3c, time-weighted mean (TWM) mAb concentrations were calculated from the ELISA data by using the trapezoidal rule to find area under the curve between each assay timepoint and dividing by the time intervals; in the absence of measured peak (day 0) levels, a constant rate of decay from day 0 through the measured values at days 3 and 7 was assumed.

Assays of GIA were performed using the FVO parasite line[1]. Assays using purified mAb to assess function after chimerization (results in Fig. 1a) were performed in the Jenner Institute Laboratory. Assays using *Aotus* samples (results in Fig. 3e) were performed at the GIA Reference Center Laboratory, NIAID, NIH. IgG was purified from pooled plasma samples of 100 μL from each of a pair of monkeys (within the same mAb treatment group), collected on day 10 after challenge. Both laboratories used the GIA Reference Centre's method, involving single-cycle parasite growth and colorimetric detection of parasite lactate dehydrogenase[1].

GIA EC₅₀ was estimated by linear interpolation. GIA₅₀ titer was calculated by dividing the day 0 plasma total IgG concentration by the total IgG GIA EC₅₀.

**Reporting summary.** Further information on research design is available in the Nature Research Reporting Summary linked to this article.

## Data analysis and availability of data and reagents

All analyses were performed using Prism software (GraphPad). All relevant data is presented within the manuscript in graphical form. Raw numerical data and reagents are available from the authors on reasonable request.

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

## Acknowledgements

We are grateful for assistance with Ebola reagents from Sean Elias, Catherine Cherry, and Geneviève Labbé; for permission to use the hIgG1Δnab Fc region, advice and access to unpublished data from Mike Clark and Kathryn Armour; and for assistance with contracts from Fay Nugent and Carly Banner. The study was supported in part by the University Challenge Seed Fund (Oxford University Innovation); A.D.D. is supported by the Wellcome Trust [grant 201477/Z/16/Z] and is a Jenner Investigator; the GIA work was supported by the United States Agency for International Development (USAID) and the Intramural Program of the National Institutes of Health, National Institute of Allergy and Infectious Diseases; DGWA held a UK Medical Research Council (MRC) iCASE PhD Studentship [MR/K017632/1]; and S.J.D. is a Jenner Investigator; a Lister Institute Research Prize Fellow and a Wellcome Trust Senior Fellow [106917/Z/15/Z]. Z.A.Z. and G.J.W. were supported by the Wellcome Trust grant 206194. The funders had no role in study design, data collection and analysis, decision to publish, or preparation of the paper.

## Author contributions

A.D.D., G.C.B., J.J., K.M., A.D., Z.A.Z., J.A.V., S.E.S., J.M.M., D.G.W.A., C.W., N.J.E., K.P.L., L.A.G.-P., C.M.L., and J.M.R. performed the experiments. A.D.D., G.C.B., J.J., K.M., A.D., J.M.M., C.W. and N.J.E. analyzed the data. ADD and SJD conceived the study. ADD, GJW, CAL and SJD obtained funding. A.D.D., G.C.B., K.M., C.M.L., G.J.W., C.A.L., J.M.R. and S.J.D. supervised the conduct of the study. A.D.D. and S.J.D. wrote the paper, with all authors commenting upon drafts.

## Additional information

**Competing interests:** A.D.D., D.G.W.A., G.J.W., and S.J.D. are named inventors on patents relating to use of PfRH5 vaccines and anti-PfRH5 antibodies for prevention or treatment of malaria. The remaining authors declare no competing interests.

**Disclaimer:** The views expressed in this article are those of the authors and do not necessarily reflect the official policy or position of the Department of the Navy, Department of Defense, nor the U.S. Government.

**Copyright statement:** Some authors (G.C.B., C.M.L., and J.M.R.) are military service members (or employees of the U.S. Government). This work was prepared as part of their official duties. Title 17 U.S.C. 105 provides that "Copyright protection under this title is not available for any work of the United States Government". Title 17 U.S.C. 101 defines U.S. Government work as a work prepared by a military service member or employee of the U.S. Government as part of that person's official duties.

