## [Peer Review File · Nature Communications]

REVIEWERS' COMMENTS:

Reviewer #1 (Remarks to the Author):

PfRh5 is one of the leading vaccine candidates under development for *P. falciparum* blood stages. This manuscript by Douglas et al uses a non-human primate model to understand if there is a causal relationship between protection from infection and the use of antibodies that only function through inhibition of PfRh5. The antibodies used in this study have been well-characterized previously to inhibit *P. falciparum* growth in culture. The Fc region of the anti-PfRh5 antibodies have been modified to contain mutations in regions that are critical for activation human FcRs, allowing the authors to focus on the antibody properties of neutralization vs other immune effector functions. Characterization of these modified antibodies show that they do not bind Aotus C1q and do not have any measurable interactions with Aotus Fcγ receptors as compared to an Ebola mAb with a wildtype Fc region.

It was hard to find anything substantially incorrect or inadequate about this paper. The figures are well-presented and the legends provide all the relevant information. The conclusions are well balanced and do not overstate the protection provided by passive transfer of anti-PfRh5 antibodies, in light of the high levels of antibodies that are actually required to achieve long-lasting protection.

My minor recommendations are:

1. In Figure 2, can the authors please clarify if the EBL040 being used is the EBL040-hIgG1Δnab or wildtype EBL040. If it is EBL040- hIgG1Δnab as mentioned in Figure 2A legend, please change all labels in Figure 2 to reflect this.
2. In supplemental materials, would it be useful to provide some indication of the purity of the antibodies used in this study, either by SDS-PAGE or purification readout?
3. The authors should be commended for producing the Aotus Fcγ receptors, would it be possible to also provide an SDS-PAGE readout of the final purification

Reviewer #2 (Remarks to the Author):

Douglas et al have studied the immune response induced by vaccination in monkeys of a recombinant protein of PfRH5 against *P. falciparum*. This is a very designed, performed and analyzed study. The authors have to be commended for sticking to their results not overselling them.

Minor comments

1. Please format the references to the journal style. Some references are even incomplete.
2. Statistical analyses:
 - a- Missing for figure 1A, B
 - b- Figure 2D: this test is inadequate; 3 sets of data are compared in the same figure. The authors cannot use t-test. They should use Anova (if the data follow a normal distribution; data to be provided) or Kruskal-Wallis (if they don't follow a normal distribution)
The test used and their results should be integrated in the figure legends.

Reviewer #3 (Remarks to the Author):

PfRH5 is a leading blood-stage malaria vaccine candidate because anti-PfRH5 antibodies inhibit parasite growth, and the interaction with its erythrocyte receptor basigin is essential for merozoite invasion. Moreover, the Authors have previously demonstrated a strong association between

protection in non-human primates (NHP) against PfRH5 vaccine, and merozoite-neutralizing antibody responses were measured in the growth inhibitory activity assay (GIA). However, causality of protection by this mechanism is remained unknown. In this study, they produced merozoite-neutralizing and non-neutralizing anti-PfRH5 chimeric monoclonal antibodies (mAbs) and performed a passive transfer-P. falciparum challenge study in Aotus nancymae monkeys. At the highest dose tested, all the animals given the neutralizing PfRH5-binding mAb c2AC7 survived the challenge without treatment. The results address the long-standing controversy regarding whether merozoite neutralizing antibody can cause protection, validating one vaccine-inducible immunological route to protection against the P. falciparum blood-stage but also highlighting the quantitative challenge of achieving such protection. This study is conducted by world experts in this area and the results are very important for the future malaria blood-stage vaccine development. All the works are carefully designed, clearly presented, and the manuscript is well written. However, I have several comments to improve the manuscript.

Comments

1) Lnes 95-98: Epitopes recognized by 2AC7 or 4BA7

Please add brief information about the epitopes recognized by 2AC7 & 4BA7 on PfRH5.

2) Line 103: "(and personal communication from M Clark and K Armour)"

Does this phrase essential? If not, please remove.

3) Lines 116-117: "Initial doses were administered immediately after challenge"

Why after the challenge? From the vaccine study point of view, I prefer passive-immunize mAb before the challenge. Please add description why they take this approach.

In addition, please add the reason why they administer 2 more mAb injections on days 7 & 14.

Why need 2 but one?

4) Line 147 & Lines 227-229: "c2AC7 concentration in animals receiving 100 mg/kg was 728 µg/mL"

It is very interesting to compare the PfRH5 specific mAb concentration in NHP with that in naturally exposed individuals in endemic regions. If possible, please add this in the discussion section.

5) Lines 170-181: "'GIA50', calculated by dividing the plasma total IgG concentration by the total IgG GIA EC50 – thus giving the plasma dilution factor required to achieve 50 % GIA in the assay." Please explain WHY the 'GIA50' parameter needs to be introduced? Without any figure, difficult to understand for the general readers of this journal.

6) Lines 195-197: "Our present study complements and extends that finding in an intact immunocompetent NHP model known to support P. falciparum sequestration."

Does "FVO-Aotus nancymae" model itself also confirmed to sequester (no schizont in the peripheral circulation)?

7) Lines 215-216: "instead our data encourage assessment of the in vivo protective capacity of other candidate mAbs selected for their ability to achieve non-GIA activities in vitro."

Does this mean that any candidates (i.e. MSP3) without GIA activity by IgG itself is worthwhile to test as this study if mAb is available? Please clarify the description.

8) Line 249: "will be reported elsewhere (Rijal P et al., in preparation)."

"Rijal P et al., personal communication." sounds better.

9) Lines 325-326: "ii) Hct fell to $\leq 25\%$ "

Does this criterion mean whether "the monkey health condition is bad not because of malaria" or "progressive parasite growth"? Please clarify for the general readers in this journal.

REVIEWERS' COMMENTS:

Reviewer #1 (Remarks to the Author):

PfRh5 is one of the leading vaccine candidates under development for P. falciparum blood stages. This manuscript by Douglas et al uses a non-human primate model to understand if there is a causal relationship between protection from infection and the use of antibodies that only function through inhibition of PfRh5. The antibodies used in this study have been well-characterized previously to inhibit P. falciparum growth in culture. The Fc region of the anti-PfRh5 antibodies have been modified to contain mutations in regions that are critical for activation human FcRs, allowing the authors to focus on the antibody properties of neutralization vs other immune effector functions. Characterization of these modified antibodies show that they do not bind Aotus C1q and do not have any measurable interactions with Aotus Fcy receptors as compared to an Ebola mAb with a wildtype Fc region.

It was hard to find anything substantially incorrect or inadequate about this paper. The figures are well-presented and the legends provide all the relevant information. The conclusions are well balanced and do not overstate the protection provided by passive transfer of anti-PfRh5 antibodies, in light of the high levels of antibodies that are actually required to achieve long-lasting protection.

We thank the reviewer for these positive comments.

My minor recommendations are:

1. In Figure 2, can the authors please clarify if the EBL040 being used is the EBL040-hlgG1Δnab or wildtype EBL040. If it is EBL040- hlgG1Δnab as mentioned in Figure 2A legend, please change all labels in Figure 2 to reflect this.

We apologise for any ambiguity, but only the EBL040-hlgG1Δnab antibody was used *in vivo*. We have now clarified this in the labelling of Figures 2A, 2D, and 3A.

2. In supplemental materials, would it be useful to provide some indication of the purity of the antibodies used in this study, either by SDS-PAGE or purification readout?

We have added a statement to the methods section which describes the production of the mAbs, making clear that "SEC chromatograms confirmed >99.5% purity (in that >99.5% of the mAb preps were of the expected molecular weight)."

3. The authors should be commended for producing the Aotus Fcy receptors, would it be possible to also provide an SDS-PAGE readout of the final purification

The Fc receptors were expressed with biotin acceptor peptide tags and co-transfected BirA biotin ligase, resulting in enzymatically monobiotinylated proteins in the culture supernatant. Using the Biacore streptavidin CAP chip, the receptors were then captured without purification. It is therefore not possible to provide an SDS-PAGE, but the capture is highly specific because of the specificity of the biotin-streptavidin interaction.

Reviewer #2 (Remarks to the Author):

Douglas et al have studied the immune response induced by vaccination in monkeys of a recombinant protein of PfrH5 against P. falciparum. This is a very designed, performed and analyzed study. The authors have to be commended for sticking to their results not overselling them.

Thank you for these comments.

Minor comments

1. Please format the references to the journal style. Some references are even incomplete.

We have reviewed the reference list and believe that all are now complete and in conformity with the journal style.

2. Statistical analyses:

a- Missing for figure 1A, B

Figure 1A shows that the GIA activity of c2AC7 is not substantially different from that of 2AC7 (and that c4BA7, like 4BA7, does not have substantial GIA activity). Figure 1B shows that mAbs bearing the hlgG1Δnab Fc region do not detectably bind Aotus C1q (remaining below the detection threshold shown; EBL040-hlgG1 is simply a positive control).

Within these figures, we are not seeking to compare conditions in order to conclude that there is a statistically or biologically significant difference between them, and therefore we do not believe that any statistical analysis is necessary or appropriate.

b- Figure 2D: this test is inadequate; 3 sets of data are compared in the same figure. The authors cannot use t-test. They should use Anova (if the data follow a normal distribution; data to be provided) or Kruskal-Wallis (if they don't follow a normal distribution) The test used and their results should be integrated in the figure legends.

We believe that a t-test (with Welch's correction for unequal variance) is appropriate here, and is the primary analysis pre-specified in the protocol which underwent scientific and ethical review.

The question being asked is not 'are any of these three groups different from the mean?' (which is the question 'asked' by an ANOVA or Kruskal Wallis, notwithstanding the possibility of performing pairwise post-tests). Instead, the question of main interest is specifically 'is there a difference between group A [EBL040 control] and group C [c2AC7 treatment]?', with the result in the c4BA7 group being of secondary interest. The desire to make a primary comparison between these two groups was foreseen during design of the study and hence the t-test was pre-specified, and was the basis for the calculation of statistical power and hence the study group sizes.

The choice of test does not affect the statistical significance of the results – the difference between group A and group C would be highly statistically significant with any of the above tests.

Reviewer #3 (Remarks to the Author):

PfRH5 is a leading blood-stage malaria vaccine candidate because anti-PfRH5 antibodies inhibit parasite growth, and the interaction with its erythrocyte receptor basigin is essential for merozoite invasion. Moreover, the Authors have previously demonstrated a strong association between protection in non-human primates (NHP) against PfRH5 vaccine, and merozoite-neutralizing antibody responses were measured in the growth inhibitory activity assay (GIA). However, causality of protection by this mechanism is remained unknown. In this study, they produced merozoite-neutralizing and non-neutralizing anti-PfRH5 chimeric monoclonal antibodies (mAbs) and performed a passive transfer-P. falciparum challenge study in Aotus nancymae monkeys. At the highest dose tested, all the animals given the neutralizing PfRH5-binding mAb c2AC7 survived the challenge without treatment. The results address the long-standing controversy regarding whether merozoite neutralizing antibody can cause protection, validating one vaccine-inducible immunological route to protection against the P. falciparum blood-stage but also highlighting the quantitative challenge of achieving such protection. This study is conducted by world experts in this area and the results are very important for the future malaria blood-stage vaccine development. All the works are carefully designed, clearly presented, and the manuscript is well written. However, I have several comments to improve the manuscript.

Thank you for these comments and helpful suggestions, which we have sought to address.

Comments

1) Lnes 95-98: Epitopes recognized by 2AC7 or 4BA7

Please add brief information about the epitopes recognized by 2AC7 & 4BA7 on PfRH5.

We have added the requested information at lines 69-72, with some re-wording of the preceding/following sentences to accommodate this.

2) Line 103: “(and personal communication from M Clark and K Armour)”. Does this phrase essential? If not, please remove.

This phrase refers to the unpublished data provided in our original cover letter showing that the hlgG1Δnab Fc region does not activate human complement. There is to our knowledge no published data showing this, and so we think this is necessary.

3) Lines 116-117: “Initial doses were administered immediately after challenge” Why after the challenge? From the vaccine study point of view, I prefer passive-immunize mAb before the challenge. Please add description why they take this approach. In addition, please add the reason why they administer 2 more mAb injections on days 7 & 14. Why need 2 but one?

For ethical reasons, we sought to minimise the number of times the animals were handled, anaesthetised and cannulated, and hence to administer the challenge and initial mAb dose through a single cannula during a single episode of anaesthesia. The reason for the choice to administer challenge and then mAb was in order to avoid exposing the challenge inoculum to the possibility of a high local concentration of mAb within the cannula used for administration. We felt that this was a theoretical risk if the challenge was given second, despite flushing.

Regarding the second part of this comment, we have re-worded the methods to clarify that a total of two additional doses were given, one on day 7 and one on day 14.

4) Line 147 & Lines 227-229: “c2AC7 concentration in animals receiving 100 mg/kg was 728 µg/mL”

It is very interesting to compare the PfrH5 specific mAb concentration in NHP with that in naturally exposed individuals in endemic regions. If possible, please add this in the discussion section.

We thank the reviewer for this helpful suggestion and have added discussion of this at lines 201-206. Anti-PfrH5 antibody concentrations in individuals living in malaria-endemic regions are orders of magnitude lower than those reached in the *Aotus* in the current study.

5) Lines 170-181: “‘GIA50’, calculated by dividing the plasma total IgG concentration by the total IgG GIA EC50 – thus giving the plasma dilution factor required to achieve 50 % GIA in the assay.” Please explain WHY the ‘GIA50’ parameter needs to be introduced? Without any figure, difficult to understand for the general readers of this journal.

We introduced the GIA50 parameter in our previous *Aotus* study¹. It provides a single parameter to reflect the level of GIA activity in any sample, and is consistent with the common practice of reporting neutralizing antibody titers against other pathogens.

Reporting GIA at a single tested IgG concentration fails to discriminate between samples achieving > 80% or < 20% GIA at that concentration. Reporting only the total IgG GIA EC50 fails to take account of the fact that IgG concentrations can vary widely between different animals, or within an individual at different timepoints¹.

Imagine 50% GIA is achieved in a test well containing 2mg/mL total IgG from animal ‘A’, blood from which has 2 mg of IgG per mL of plasma- this reflects an GIA50 titer of 1. If 50% GIA is *also* achieved in a test well containing 2mg/mL total IgG from animal ‘B’, but in contrast blood from ‘B’ has 10 mg of IgG per mL of plasma, this reflects an GIA50 titer of 5. It is reasonable to expect that parasites exposed to animal ‘B’s’ plasma would be exposed to a greater ‘force’ of neutralizing antibody.

6) Lines 195-197: “Our present study complements and extends that finding in an intact immunocompetent NHP model known to support *P. falciparum* sequestration.” Does “FVO-*Aotus nancymae*” model itself also confirmed to sequester (no schizont in the peripheral circulation)?

This is correct. In our own studies with *P. falciparum* FVO in *A. nancymae*, schizonts are rare in the peripheral circulation- they are present at c. 5% of the number of circulating ring-stage parasites, indicating >90% sequestration of schizonts.

Although using a different parasite strain (*P. falciparum* M. Camp) and a different monkey species within the *Aotus* genus (*A. trivargatus*, closely related to *A. nancymae*), the paper we cite regarding *P. falciparum* sequestration in *Aotus* was seminal in providing understanding of this important mechanism of pathogenesis².

7) Lines 215-216: “instead our data encourage assessment of the in vivo protective capacity of other candidate mAbs selected for their ability to achieve non-GIA activities in vitro.” Does this mean that any candidates (i.e. MSP3) without GIA activity by IgG itself is worthwhile to test as this study if mAb is available? Please clarify the description.

The intended meaning is exactly as the reviewer suggests, but we have re-worded this section of the discussion to provide greater clarity.

8) Line 249: “will be reported elsewhere (Rijal P et al., in preparation).” “Rijal P et al., personal communication.” sounds better.

We have worded this as ‘in preparation’, on the basis that there is substantial overlap between the authorship of that manuscript and the current study, including the same senior author.

9) Lines 325-326: “(ii) Hct fell to $\leq 25\%$ ” Does this criterion mean whether “the monkey health condition is bad not because of malaria” or “progressive parasite growth”? Please clarify for the general readers in this journal.

We have added a statement at this point to clarify that severe anemia is a common complication of poorly controlled malaria infection in *Aotus*. Strictly speaking, although this *is* directly due to the malaria infection, this is not the same as ‘progressive parasite growth’ – anaemia often occurs after the peak of parasitaemia, during the phase of immunological control of parasite replication.

REFERENCES

1. Douglas, A.D., *et al.* A PfRH5-based vaccine is efficacious against heterologous strain blood-stage *Plasmodium falciparum* infection in *Aotus* monkeys. *Cell Host & Microbe* **17**, 130-139 (2015).
2. Miller, L.H. Distribution of mature trophozoites and schizonts of *Plasmodium falciparum* in the organs of *Aotus trivirgatus*, the night monkey. *Am J Trop Med Hyg* **18**, 860-865 (1969).